

# Genome size versus geographic range size in birds

Beata Grzywacz[1] and Piotr Skórka[2]

[1] Institute of Systematics and Evolution of Animals, Polish Academy of Sciences, Kraków, Poland
[2] Institute of Nature Conservation, Polish Academy of Sciences, Kraków, Poland

## ABSTRACT

Why do some species occur in small, restricted areas, while others are distributed globally? Environmental heterogeneity increases with area and so does the number of species. Hence, diverse biotic and abiotic conditions across large ranges may lead to specific adaptations that are often linked to a species' genome size and chromosome number. Therefore, a positive association between genome size and geographic range is anticipated. Moreover, high cognitive ability in organisms would be favored by natural selection to cope with the dynamic conditions within large geographic ranges. Here, we tested these hypotheses in birds—the most mobile terrestrial vertebrates— and accounted for the effects of various confounding variables, such as body mass, relative brain mass, and geographic latitude. Using phylogenetic generalized least squares and phylogenetic confirmatory path analysis, we demonstrated that range size is positively associated with bird genome size but probably not with chromosome number. Moreover, relative brain mass had no effect on range size, whereas body mass had a possible weak and negative effect, and range size was larger at higher geographic latitudes. However, our models did not fully explain the overall variation in range size. Hence, natural selection may impose larger genomes in birds with larger geographic ranges, although there may be additional explanations for this phenomenon.

## INTRODUCTION

There is enormous variation in the sizes of species' geographic ranges (*Gaston, 2003*). There are several explanations for this, with leading hypotheses invoking traits such as body size (*Cambefort, 1994*; *Gaston & Blackburn, 2000*), dispersal ability (*Lester et al., 2007*; *Laube et al., 2013*), and niche breadth (*Garcia-Barros & Romo Benito, 2010*). The increase in geographic range size is consistent with environmental variability (for example, climate), which is considered a major selective evolutionary force (*Lee-Yaw & Irwin, 2012*; *Sayol et al., 2016*; *Liedtke et al., 2018*). Therefore, species possess numerous traits for living in dynamic environments (*Gaston & Blackburn, 2000*; *Zamudio, Bell & Mason, 2016*). These traits have strong heritable components and are thus linked with gene number (*Zhang et al., 2014*). Hence, genome size could be an important predictor of a species' range size and could affect other species' traits subject to natural selection.

The evolution of genome size is multifaceted (*Lefébure et al., 2017*). According to the "selection hypothesis", the variation in genome size has consequences on organismal fitness

Corresponding author
Piotr Skórka, pskorka@iop.krakow.pl

and may thus be subject to selection (*Gregory & Hebert, 1999*; *Petrov, 2001*). Corroborating this hypothesis, *Hou & Lin (2009)* found a strong positive association between the log-transformed values of protein-coding gene number and genome size in eukaryotes and non-eukaryotes. In eukaryotes, genome size may be defined as the C-value, which is the amount of DNA per haploid genome and chromosome number. Genome size regulates the cell size associated with polyploidy, possibly leading to instantaneous shifts in the physiological tolerance and trait values (*Levin, 2002*). Alternatively, according to the "junk DNA" hypothesis, the propagation of selfish intragenomic transposons and other mobile genetic elements leads to the accumulation of mutations throughout the genome, yielding a larger genome size (*John & Miklos, 1988*; *Bennetzen & Kellogg, 1997*). Indeed, organisms with larger genomes tend to have longer introns and more transposable elements than organisms with smaller genomes (*Lynch & Conery, 2003*; *Charlesworth & Barton, 2004*).

These two hypotheses are often combined by postulating adaptive functions of this additional DNA, given that DNA abundance, rather than its contents, produces a direct and significant effect on the phenotype (*Petrov, 2001*). For instance, a larger genome size may be an adaptive strategy, because it may directly or indirectly increase the nuclear and cellular volume (*Cavalier-Smith, 1978*) and body size (*Gregory, 2005*), buffer fluctuations in the enzyme concentrations, or protect the coding DNA from mutations (*Hsu, 1975*; *Janssen, Colmenares & Karpen, 2018*). Likewise, genome size is correlated with cell cycle complexity (*Gregory, 2002*; *Yu et al., 2019*), basal metabolism (*Vinogradov, 1997*), tissue differentiation, and developmental rate (*Sessions & Larson, 1987*; *Xia, 1995*; *Wyngaard et al., 2005*). *Arnqvist et al. (2015)* showed that females with larger genomes laid more eggs and males with larger genomes fertilized more eggs in beetles.

The hypothesis regarding genome size versus geographic range size has already been tested in bacteria and plants. Bacteria with larger genomes are more likely to have wider environmental and geographic ranges than those with smaller genomes (*Barberán et al., 2014*; *Choudoir et al., 2018*). In contrast, while plant invasiveness is negatively associated with genome size but positively associated with chromosome number (and ploidy level), plant genome size is positively associated with chromosome number (*Pandit, White & Pocock, 2014*). However, this hypothesis has not been tested in vertebrates.

In contrast to plants, genome size (C-value) may be weakly but positively correlated with chromosome number in animals (*Vinogradov, 1998*; *Elliott & Gregory, 2015*). Chromosome number plays pivotal roles in speciation, sex determination, and developmental modes (*King, 1995*; *Warchałowska-Śliwa et al., 2011*; *Blackmon, Ross & Bachtrog, 2017*; *Lucek, 2018*). Thus, chromosome number may also be positively associated with geographic range size (*Guo, Kato & Ricklefs, 2003*; *Martinez et al., 2017*).

Birds are a unique and useful model group to test many evolutionary hypotheses. They have a limited genome size compared to other vertebrates, ranging from 1.15 to 1.62 pg of DNA per haploid genome (*Andrews, Mackenzie & Gregory, 2009*). However, there is substantial variation in avian karyotypes; as such, the chromosomes are further divided into macro- and microchromosomes (*Kretschmer, Ferguson-Smith & De Oliveira, 2018*; *Degrandi et al., 2020*). Avian genomic diversity covaries with adaptations to different life strategies and convergent evolution of traits (*Zhang et al., 2014*). Most birds possess the

ability to fly; they are thus not as constrained by physical barriers, as are other organisms. Range size in birds is therefore often correlated with their dispersal ability (*Böhning-Gaese et al., 2006*; *Laube et al., 2013*). However, avian flight required massive changes (for light weight and increased energy efficiency) of all aspects, including the size of the genome (*Zhang et al., 2014*). In birds amount of DNA gained by transposable element expansion is counteracted by DNA loss from large segmental deletions (*Kapusta, Suh & Feschotte, 2017*; *Zhang et al., 2014*). Nevertheless, bird genome size is positively associated with the nuclear or cellular size and wing loading index, which is an indicator of adaptation for efficient flight (*Andrews, Mackenzie & Gregory, 2009*). Hence, a positive association between genome size and geographic range is anticipated in birds.

The alternative (but not mutually exclusive) hypothesis is that large geographic ranges favor enhanced cognitive skills, enabling survival in dynamic conditions across these ranges. Cognitive skills are linked to a large brain (*Reader & Laland, 2002*; *Sol et al., 2005*; *Emery, 2006*) and seemingly to habitat generalism (*Edmunds, Laberge & McCann, 2016*; *Navarrete et al., 2016*). Indeed, bird species exposed to greater environmental variation throughout their geographic range are likely to have larger brains (*Sol et al., 2005*; *Sayol et al., 2016*). Therefore, geographic range size may be positively correlated with brain size. In addition, brain size is strongly correlated with body size (*Minias & Podlaszczuk, 2017*). Thus, body size should always be considered a covariate in range size and genome studies, because there is a well-documented paradigm of overall positive association between body size and range size in animals (*Gaston & Blackburn, 2000*; *Newsome et al., 2019*).

Most bird lineages have diversified within rather restricted regions, and many tropical species are highly reluctant to cross unfamiliar habitats despite being able to fly (*Gillies & St. Clair, 2010*). In terms of their adaptability to a broad range of climates, it is often assumed that birds are constrained by niche conservatism, which appears to be asymmetrical. Ancient tropical groups cannot easily adapt to, or expand into, cold climates; however, groups that have evolved at high latitudes and are cold tolerant are actually thermally flexible and can easily adapt to new climates. They are therefore often the founders of breeding populations (and species proliferation) within the tropics (*Smith et al., 2012*; *Khaliq et al., 2015*; *Winger et al., 2019*). Thus, the range size of birds is expected to be larger at higher latitudes.

To this end, in this study, we tested the hypothesis that genome size, chromosome number, relative brain size, body size and latitude are positively associated with geographic range size in birds. Efficient testing of the causality of such associations on a broad taxonomic scale has proven difficult in the past due to the intercorrelations and phylogenetic non-independence of these biological traits. Thus, we used phylogenetic generalized least squares (PGLS) and phylogenetic confirmatory path analysis (PPA) to control for phylogeny and evolutionary constraints while accounting for the multicollinearity of variables.

## METHODS

### Data collection

Data on bird species' ranges were collected from *BirdLife International (2019)* (http://datazone.birdlife.org/species/requestdis). Data were manipulated in QGIS 3 Noosa (*QGIS*

*Development Team, 2019*). To calculate range areas, the Bonne equal-area transformation (ESRI: 54024) was applied. Only extant native ranges were used (*Ravilious et al., 2015*). Bird phylogenetic trees (*Jetz et al., 2012*; *Jetz et al., 2014*) based on the constraints described by *Hackett et al. (2008)* were created in a nexus format online (http://birdtree.org/). Sets of 1,000 trees were downloaded for three data subsets (see below).

The method used by *Jetz et al. (2012)* and *Jetz et al. (2014)* allows the inclusion of taxa for which there are no real-time data; this can yield some very problematic results (*Hosner, Braun & Kimball, 2015*; *Wang et al., 2017*). However, newer avian megaphylogenies are available (*Ksepka et al., 2020*). The tree created by *Ksepka et al. (2020)* is based on the reanalysis of the supermatrix described by *Burleigh & Kimball (2015)*. *Ksepka et al. (2020)* used constraints from the tree reported by *Jarvis et al. (2014)*, which reflects analyses of approximately 40 Mbp of aligned data and includes over 10,000 loci. The tree created by *Ksepka et al. (2020)* includes fewer species than that created by *Jetz et al. (2012)*; *Jetz et al. (2014)*; however, we used a subset of species present in both to assess whether the results of statistical analyses differ between the two trees.

Data on species genome size were compiled from the Animal Genome Size Database (http://www.genomesize.com/search.php). This database contains both C-value and chromosome number data. We gathered C-value data for 637 bird species. To complete chromosome number data, which was extremely limited, we also used data published by *Kretschmer, Ferguson-Smith & De Oliveira (2018)*. Body mass data were obtained from *Wilman et al. (2014)*. Brain mass data were compiled from three published sources (*Fristoe, Iwaniuk & Botero, 2017*; *Minias & Podlaszczuk, 2017*; *Tsuboi et al., 2018*). Overall, both brain size and genome size data were available for 311 species. Finally, genome size, chromosome number, brain size, and body size data were available for 65 species. All data used in the analyses are available in Supplementary Material 1. Phylogenetic trees used in analyses (see below) are attached as Supplmentary Material 2 and 3. Bird order and species names follow *Gill, Donsker & Rasmussen (2020)*.

## Data analysis

We computed three single-consensus trees using stringent consensus methods implemented in PAUP* 4.0 (*Swofford, 2003*) from the three sets of 1,000 trees published by *Jetz et al. (2012)* and *Jetz et al. (2014)* downloaded from birdtree.org. The consensus methods used show the tree group sequences only if that grouping appears in all trees in the set. Three consensus trees were used in subsequent statistical analyses. The tree branch lengths were generated from the fitted branch lengths of the 1,000 input trees using the "consensus.edges" function of the phytools package in R (*Revell, 2012*). The trees were visualized with Iroki (*Moore et al., 2020*) using the ggplot package in R (*Wickham, 2016*), as shown in Fig. S1. Moreover, the summary statistics on geographic range size, C-value and chromosome number were visualized in Fig. 1 using a phylogenetic tree from *Kimball et al. (2019)*. The species for which range size centroid is located below 30° latitude are considered "tropical" species and are depicted in a different color from other species in visualizations.

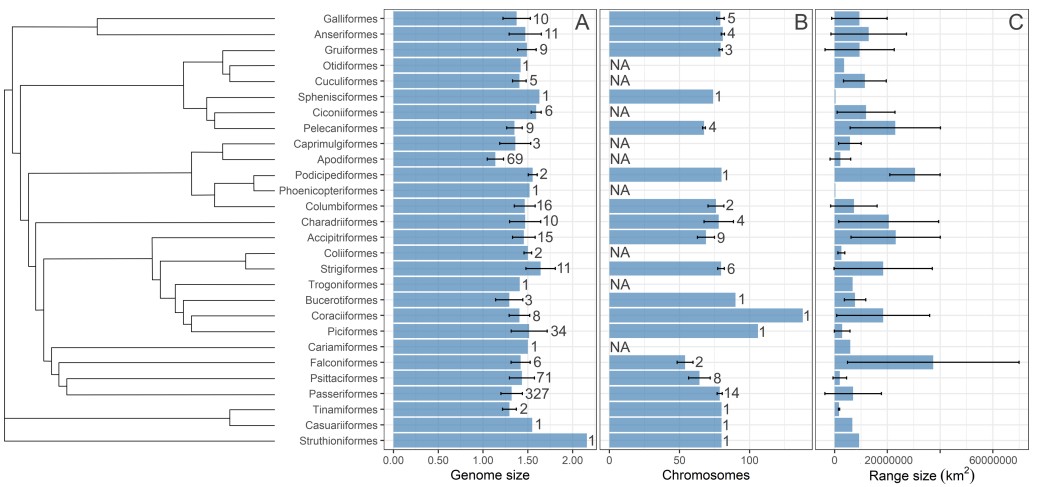

**Figure 1** **A summary statistics of the studied variables.** The distirubution of bird (A) genome size, (B) number of chromosomes, and (C) geographic range size on the phylogenetic tree. The phylogenetic tree is from *Kimball et al. (2019)*. Bars are means with standard deviations. Sample size (number of species) is given next to each bar. NA indicates that data was not available. Sample size for the geographic range size is the same as for the genome size.

We performed two sets of analyses: one based on the consensus trees published by *Jetz et al. (2012)* and *Jetz et al. (2014)* and another based on the tree published by *Ksepka et al. (2020)*. We used PGLS models implemented in the nlme package (*Von Hardenberg & Gonzalez-Voyer, 2013*; *Pinheiro et al., 2019*) and used the Brownian motion correlation structure of the model residuals to account for the phylogenetic dependence of species (*Münkemüller et al., 2012*). Brownian motion is a popular model in comparative biology, because it captures the potential trends of trait evolution under a reasonably wide range of scenarios (*Münkemüller et al., 2012*; *Harmon, 2019*). Range sizes of the various bird species (km$^2$) was the dependent variable, while genome size (C-value) and chromosome number were the primary explanatory variables. We also included following covariates: body mass, relative brain mass, and geographic latitude (absolute values). The relative brain mass residual in the PGLS model was not explained by body mass. These variables are well linked to environmental variability and other species traits. Body mass and relative brain mass were logarithmically transformed. The number of available data varied among the explanatory variables; therefore, we built three models based on the trees published by *Jetz et al. (2012)* and *Jetz et al. (2014)* for explaining range size variation in birds. The first model (637 species) included three explanatory variables, namely genome size, body mass and latitude. The second model (311 species) included four explanatory variables, namely genome size, body mass, relative brain mass and latitude. The third model (65 species) included five explanatory variables, namely genome size, chromosome number, body mass, relative brain mass and latitude. Each model was tested against the null model (the model with the intercept alone) using the likelihood ratio test. The Nagelkerke pseudo-R square, calculated in the companion package of R, was used as the measure of model fit (*Mangiafico, 2020*).

Furthermore, we used PPA (*Von Hardenberg & Gonzalez-Voyer, 2013*) based on prespecified candidate path models to test for the effects of traits on range variation using the phylopath package in R (*Van der Bijl, 2018*). This approach allowed us to compare the causal hypotheses regarding the associations among traits, disentangling the direct effects from the indirect ones, while correcting for the non-independence of the trait data due to common ancestry (*Santini et al., 2019*). In addition, this model accounts for trait multicollinearity (Fig. 2) better than multivariate linear models, because the variance of the response is partitioned among fewer predictors (*Gonzalez-Voyer & Von Hardenberg, 2014*). To build paths, we used data from the third model, which contained all explanatory variables. The only difference was that we used raw data on brain mass (logarithmically transformed) because the analysis enables to disentangle the complex relations among variables. A total of 22 path model combinations were built with different configurations of these variables. We used a set of hypotheses depicted by directed acyclic graphs (Fig. 3) to minimize the number of models for testing (*Gonzalez-Voyer et al., 2016*). The first set of models included the direct impact of each explanatory variable (Fig. 3), and the second set of models included indirect effects. We assumed that (1) the effect of genome size may be mediated by chromosome number, (2) the effect of body mass may be mediated by genome size, (3) the effect of body mass may be indirect via relative brain mass, and (4) the effect of geographic latitude may be mediated by body mass (*Martin, 1981*). The third set of models included more complex indirect associations (Fig. 3). Specifically, we assumed that the effect of body mass may be mediated by both genome size and chromosome number and that the effect of genome size may be mediated by chromosome number. Finally, these sets of models were tested against the null model. The sets of models were compared using the C-statistic Information Criterion corrected for small sample sizes.

All analyses were repeated on the subset of species included in the latest phylogeny published by *Ksepka et al. (2020)*. However, the sample size was lower. The first PGLS model (262 species) included three explanatory variables, namely genome size, body mass and latitude. The second model (254 species) included four explanatory variables, namely genome size, body mass, relative brain mass, and latitude. The third model (55 species) included five explanatory variables, namely genome size, chromosome number, body mass, relative brain mass and latitude. The PPA was based on the third model with 55 species.

## RESULTS

The first PGLS model showed that range size was positively associated with genome size and latitude but negatively associated with body mass (Table 1; Fig. 4). This model was statistically different from the null model ($\chi^2 = 13.048$, $P < 0.001$) and explained 4% of the total variation in range size. The second PGLS model also revealed that range size was positively associated with genome size but negatively associated with body mass, and not associated with relative brain mass and latitude (Table 1). This model was also statistically different from the null model ($\chi^2 = 11.007$, $P = 0.008$) and explained 6% of the total variation in range size. Furthermore, the third PGLS model showed that range size was associated with genome size and body mass but not with chromosome number, relative

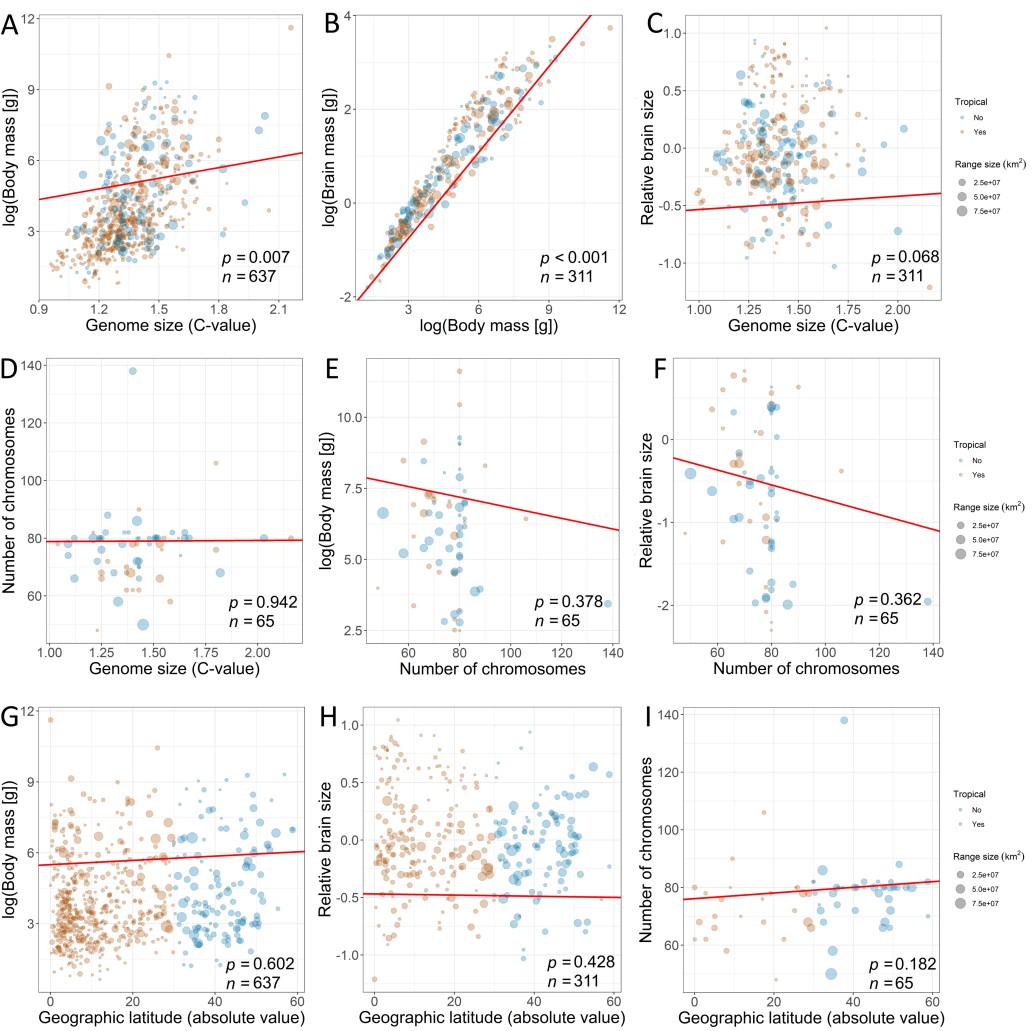

**Figure 2** **Associations among the explanatory variables revealed by phylogenetic generalized least squares (fitted red line).** The associations between (A) genome size and body mass, (B) body mass and brain mass, (C) genome size and relative brain mass, (D) genome size and number of chromosomes, (E) number of chromosomes and body mass, (F) number of chromosomes and relative brain size, (G) geographic latitude and body mass, (H) geographic latitude and relative brain mass, and (I) geographic latitude and number of chromosomes. Size of the dots is scaled according to geographic range size. The species for which geographic range centroid is located below 30 °C latitude are considered "tropical" species (orange dots) in contrast to other species (blue dots). Dots are transparent for better visibility of overlapping data. Statistical significance is presented along with sample size.

brain mass and latitude (Table 1). This model was statistically different from the null model ($\chi^2 = 12.738$, $P = 0.002$) and explained 18% of the total variation in range size. In PPA, the models with indirect simple and indirect complex effects gained the highest support (Table 2). Based on the estimated coefficients, genome size had a significant and independent effect (confidence intervals not overlapping with zero) on range size (Fig. 5).

Analysis based on the tree published by *Ksepka et al. (2020)* yielded similar results (Table S1 and Fig. S2). However, geographic latitude was significant in all models examined.
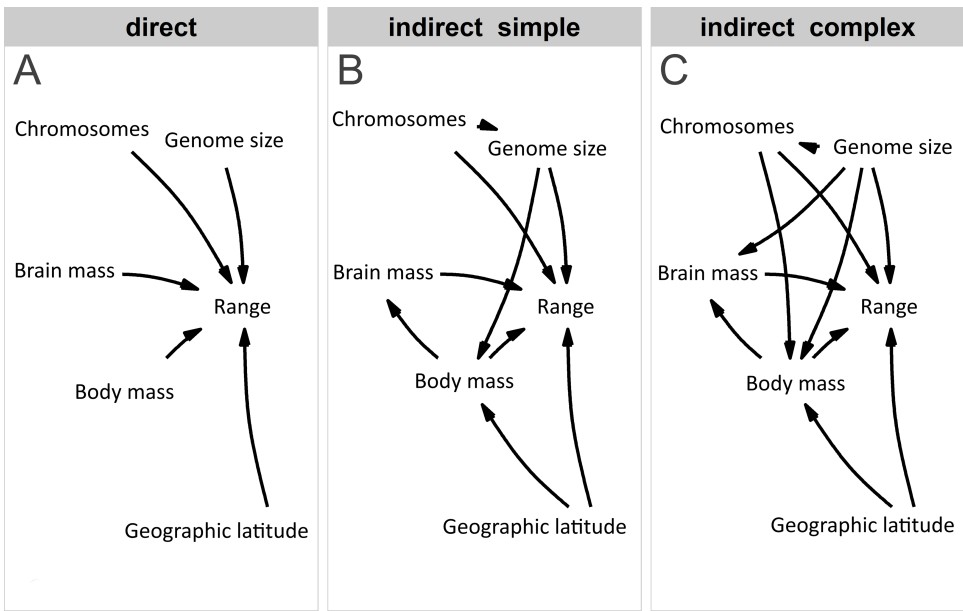

Models

**Figure 3** **Competing models in the phylogenetic confirmatory path analysis.** Competing models: (A) direct, (B) indirect simple, and (C) indirect complex for testing the associations of geographic range size (Range) with genome size, chromosome number (Chromosomes), body mass, relative brain mass and latitude in birds.

Moreover, in second model the relative brain mass had positive effect on geographic range size (Table S1). In third model the number of chromosomes had negative effect on the range size and the effect of the genome size was non-significant (Table S1). However, the third model with all explanatory variables suffered due to small sample size as revealed by running fourth model with the effect of genome size only (Table S1). Based on these limited data, PPA indicated that range size was positively associated with geographic latitude but negatively associated with chromosome number (Table S2 and Fig. S2).

## DISCUSSION

Large-scale patterns of spatial variation in species' geographic ranges are central to many fundamental questions in macroecology. However, the global nature of these patterns remains contentious. The present study confirmed our hypothesis that in birds, genome size is weakly but positively associated with geographic range size. Moreover, analysis on phylogenetic tree from *Ksepka et al. (2020)* showed that range size may be negatively correlated with chromosome numbers. Birds represent an example of a group in which genome size is correlated with active speciation. The amount of DNA gained by transposable element expansion is counteracted by DNA loss from large segmental deletions (*Kapusta, Suh & Feschotte, 2017*; *Zhang et al., 2014*). Thus, genome size regulation (*Fischer et al., 2014*) is perhaps more important to adaptation than genome size itself. Furthermore, the effect of genome size on geographic range size is not easy to predict, considering

**Table 1 Phylogenetic generalized least squares models testing association among geographic species ranges in birds and genome size, chromosome number, body mass, relative brain mass and latitude.** Three models differed by available sample size for each explanatory variable. Statistically significant effects have emboldened *P*-values.

| Model 1st (N = 637 species) | | | | |
|---|---|---|---|---|
| **Effect** | *estimate* | *SE* | *t* | *P* |
| (Intercept) | 2947223.7 | 13563675.8 | 0.217 | 0.828 |
| Genome size | 11347387.5 | 3810612.2 | 2.978 | **0.003** |
| Body mass | −1666844.5 | 792465.3 | −2.103 | **0.036** |
| Geographic latitude | 107082.4 | 34917.7 | 3.067 | **0.002** |
| **Model 2nd (N = 311)** | | | | |
| Effect | *estimate* | *SE* | *T* | *P* |
| (Intercept) | 1282122.5 | 17644249.3 | 0.073 | 0.942 |
| Genome size | 16502711.4 | 5846290.2 | 2.823 | **0.005** |
| Body mass | −2265842.6 | 1127589.4 | −2.009 | **0.045** |
| Brain mass (residual) | −1224487.9 | 5200002.4 | −0.235 | 0.814 |
| Geographic latitude | 27604.6 | 57284.6 | 0.4882 | 0.630 |
| **Model 3rd (N = 65)** | | | | |
| Effect | *estimate* | *SE* | *T* | *P* |
| (Intercept) | 25833067.5 | 31459858.8 | 0.821 | 0.415 |
| Genome size | 29848628.7 | 11134362.6 | 2.681 | **0.009** |
| Chromosome number | −345377.8 | 276973.9 | −1.247 | 0.217 |
| Body mass | −3923399.6 | 1685932.1 | −2.327 | **0.023** |
| Brain mass (residual) | 264551.6 | 8961357.9 | 0.030 | 0.976 |
| Geographic latitude | 25833067.5 | 31459858.8 | 0.821 | 0.415 |

that it largely represents the dynamic balance between positive and negative selection on genome size. According to *Lynch & Conery (2003)*, the ineffectiveness of selection in species with a low effective population size is key to genome evolution. Large organisms have lower population sizes than small ones and hence a lower effective population size. The effective population size determines whether natural selection can maintain functional DNA sequences in the face of deleterious mutations. It is almost impossible for a deleterious mutation to spread when the effective population size is large; thus, it may prevent genome enlargement. Interestingly, the positive association between population size and geographic range size is well-documented (*Gaston & Blackburn, 1996*). Thus, these contrasting forces may be the reason statistical models in this study explained only a small proportion of variation in range size. However, our results are not different from the explained variances generally reported in ecological research. According to *Møller & Jennions (2002)*, statistical models can explain between 2.5% and 5.4% of variation in ecological studies. Moreover, there may be additional explanations for the variation in range size in birds, which are mostly linked to environmental constraints, such as climate, geographic location, and habitat (*Orme et al., 2006*; *Laube et al., 2013*; *Zhang et al., 2014*; *Sayol et al., 2016*). Our results support prediction that the range size increase with latitude. This finding is in line with the Rapoport's rule which states that there is a positive latitudinal gradient in latitudinal range extent (*Rapoport, 1982*; *Stevens, 1989*). Despite there is a criticism of this

**Table 2  Results of the phylogenetic confirmatory path analysis.** Different sets of models are compared (see Fig. 3).

| Model | k | q | C | p | CICc | ΔCICc | l | w |
|---|---|---|---|---|---|---|---|---|
| indirect simple | 6 | 15 | 13.029 | 0.367 | 52.824 | 0.000 | 1.000 | 0.853 |
| indirect complex | 4 | 17 | 9.323 | 0.316 | 56.345 | 3.520 | 0.172 | 0.147 |
| direct | 10 | 11 | 93.275 | 0.000 | 120.256 | 67.431 | 0.000 | 0.000 |
| null | 15 | 6 | 113.251 | 0.000 | 126.700 | 73.875 | 0.000 | 0.000 |

Notes.
$k$, independence claims made by the model; $q$, the number of parameters; $C$, the C statistic; $p$, p-value for C; $CICc$, the C-statistic information criterion corrected for small sample sizes; $\Delta CICc$, the difference in CICc with the top model; $l$, the associated relative likelihoods; $w$, CICc weights.
A significant p indicates that the available evidence rejects the model.

rule stating that this is a local phenomenon occurring only on the northern Hemisphere above a latitude of about 40−50°N (*Ruggiero & Lawton, 1998*), there are several studies supporting the Rapoport's rule, e.g., in amphibians (*Whitton et al., 2012*), birds (*Dyers et al., 2020*) and mammals (*Arita, Rodríguez & Vázquez-Domínguez, 2005*).

Interestingly, we found that body mass was negatively correlated with range size in multivariate models. This result contradicts most previous findings, which are considered the paradigm of macroecology (*Cambefort, 1994*; *Gaston & Blackburn, 1996*; *Gaston & Blackburn, 2000*). However, *Gaston & Blackburn (1996)* used range size data of limited quality (generalized range maps); therefore, they may have greatly underestimated the range of several species with very small distributions. Hence, such an association should be investigated in more detail in additional taxa, because this knowledge may change our understanding of the role of body size in shaping range sizes. Body size depends on genome size (rather than vice versa); thus, genetic factors may be the primary causative variables, while body mass may be linked indirectly with species range.

Furthermore, we did not find effect of relative brain mass on range size, contrary to the previous reports of a positive association between environmental variation and brain size (*Sayol et al., 2016*). The effect was significant in one analysis based on *Ksepka et al. (2020)* but was not supported by PPA. Larger brains indicate higher cognitive ability (information processing) under strong selection (*Reader & Laland, 2002*; *Sol et al., 2005*). The possible explanation for this is that on a geographic scale, environmental variation may be too high for birds to adapt their cognitive skills. In this case, adaptations to dynamic conditions would evolve, which may be linked with genome size.

## Study limitations

Several issues should be taken into consideration when interpreting our results. One of the great challenges in recent studies of macroecological patterns has been how to explain the highly aggregated distribution of species with very small geographic ranges in specific tropical regions (for example, oroclines, locations near the edges of continental plates, or archipelagos (*Rahbek et al., 2007*; *Rahbek et al., 2019*; *Jønsson et al., 2017*)). Interestingly, 50% of the avian species with very small geographic ranges are exclusively found at latitudes below 30°. Unfortunately, however, the genome data for these species are poorly represented. Hence, the results depend largely on geographic sampling. Our results are

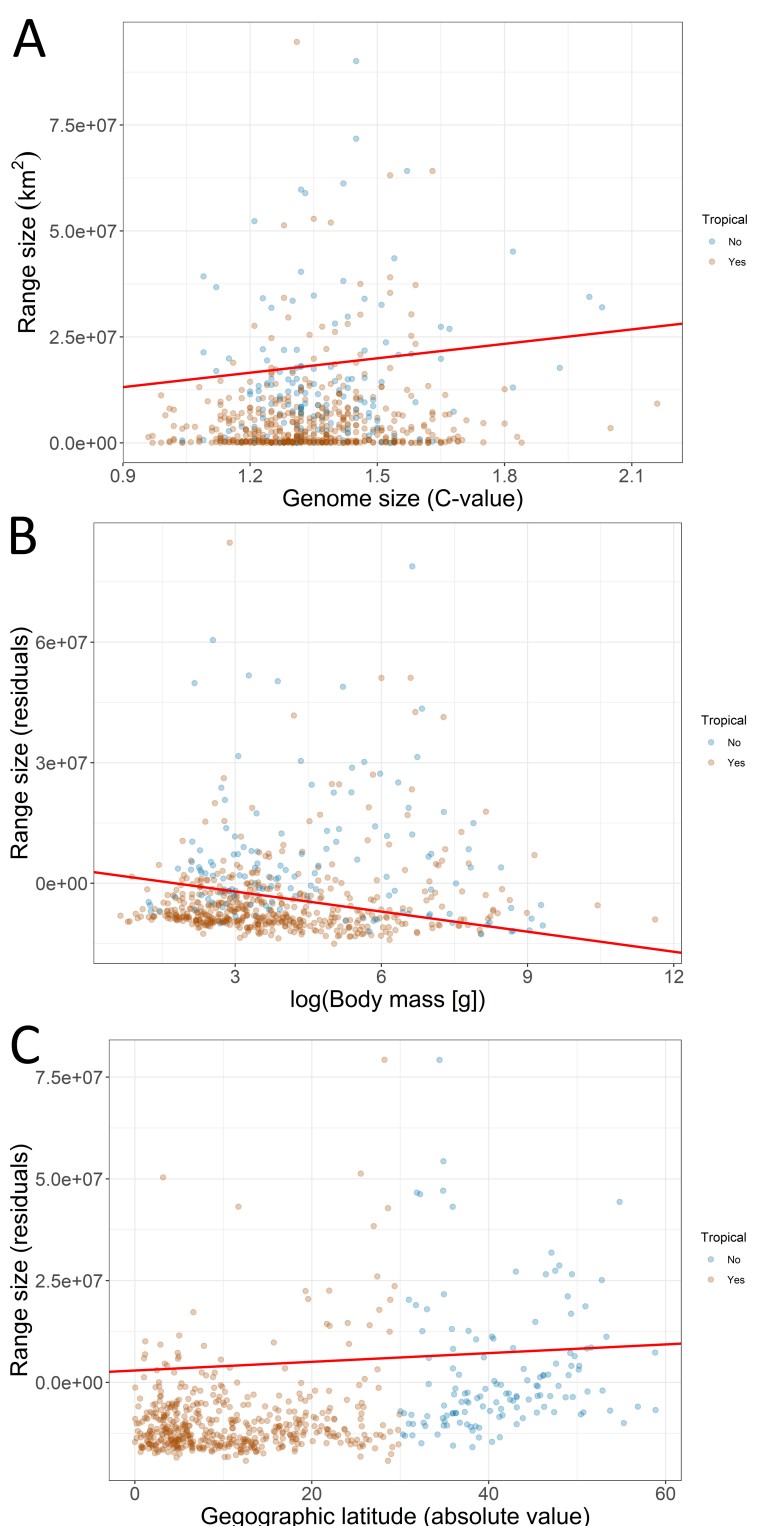

**Figure 4   Associations of geographic range size with (A) genome size, (B) body mass of birds, and (C) geographic latitude.** Fitted lines (red) are derived from the phylogenetic generalized least squares models. The effect of body mass and latitude on the residual range size (residuals not explained by genome size) is depicted. Futher explanations: see Fig. 2.

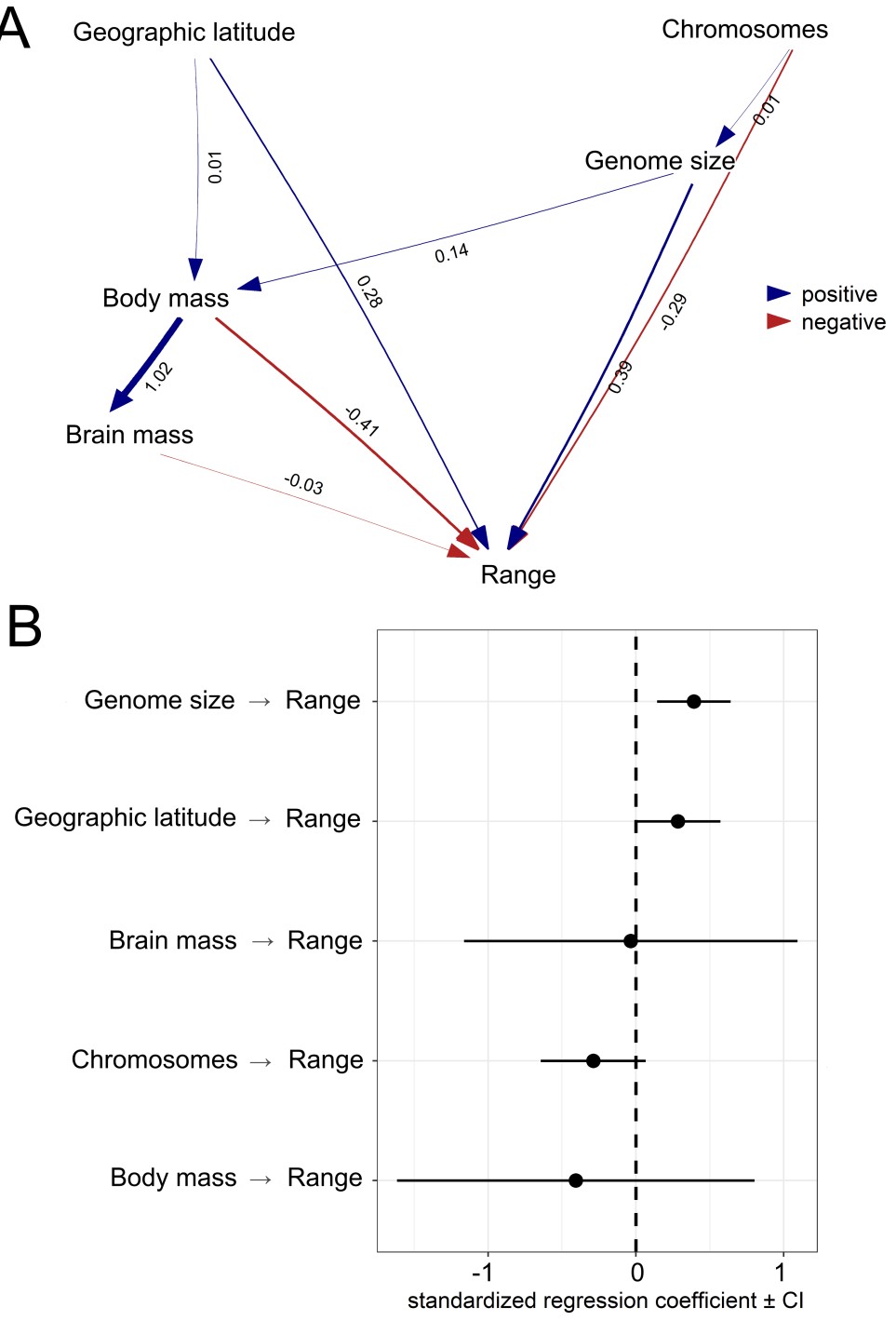

**Figure 5** **Results from the phylogenetic confirmatory path analysis.** Test supporting causal model with standardized path coefficients (A). Standardized coefficients with 95% confidence intervals (CIs) for explanatory variables associated with geographic range size of birds (B).

biased toward species with large-to-moderate geographic ranges, neglecting numerous tropical species with very small ranges. This bias may also explain the negative association between range size and body size. Another bias (or confounding factor) may be that flight ability (aerial foraging) has been found to correlate with a small genome size (*Andrews, Mackenzie & Gregory, 2009*; *Kapusta, Suh & Feschotte, 2017*). However, flight ability may also affect range size. As the genome size may be increased by the replication of transposable elements it may be also decreased by large deletions indicating that counteracting selective forces shape bird genome size (*Kapusta, Suh & Feschotte, 2017*; *Zhang et al., 2014*). Moreover, our sample size for analyses was limited by the availability of data on chromosome number. Thus, data on avian karyotypes and other traits potentially related to range should be included in future studies. Statistical methods we used assume that relationships are linear, which is not necessarily an optimal assumption (*Quader et al., 2004*). Phylogenetic comparative methods can fail to detect coevolution when the underlying relationships among traits are nonlinear (*Quader et al., 2004*). However, it is difficult to include nonlinear methods in phylogenetically corrected statistics, specifically PPA. We overcame this problem by logarithmic transformation of body mass and brain size data. Moreover, geographic latitude was included as an absolute value to allow for linear modeling.

We used two avian phylogenies. The first was proposed by *Jetz et al. (2012)* and *Jetz et al. (2014)* and included all taxa for which there are no real-time data. In that tree, there are parts of the topology for taxa with no data that have 100% support (*Hosner, Braun & Kimball, 2015*), which likely reflects the difficulty of running the Markov chain Monte Carlo algorithm long enough to adequately sample the posterior distribution when a large number of taxa are included. Meanwhile, the tree presented by *Ksepka et al. (2020)* included fewer species, omitting some avian orders (such as Casuariiformes and Ciconiiformes). Analysis based on this tree had a lower sample size and thus a lower power, particularly when building models with all explanatory variables. Analyses with both trees yielded slightly different results when the sample size was small. This indicate that further research on avian phylogeny based on genome sequences with inclusion of as many taxa as possible are required. In addition, species ranges are not constant, and the range data used have other well-known limitations, however not better options exist at that scale of study.

## ACKNOWLEDGEMENTS

We thank Alexandre Aleixo, Edward L. Braun, and Jon Fjeldså for constructive criticism on earlier versions of the manuscript.

### Funding

Beata Grzywacz was financed by the statutory funds of the Institute of Systematics and Evolution of Animals, Polish Academy of Sciences. Piotr Skórka was financed by the statutory funds of the Institute of Nature Conservation, Polish Academy of Sciences. The

funders had no role in study design, data collection and analysis, decision to publish, or preparation of the manuscript.

## Grant Disclosures

The following grant information was disclosed by the authors:
The Institute of Systematics and Evolution of Animals, Polish Academy of Sciences.
The Institute of Nature Conservation, Polish Academy of Sciences.

## Competing Interests

The authors declare there are no competing interests.

## Author Contributions

- Beata Grzywacz and Piotr Skórka conceived and designed the experiments, performed the experiments, analyzed the data, prepared figures and/or tables, authored or reviewed drafts of the paper, and approved the final draft.

## Data Availability

Raw data on birds' genome size, chromosome number, brain size, body size, latitude and geographic range size as well as the phylogenetic trees used in analyses are available in the Supplemental Files.

## Supplemental Information

Supplemental information for this article can be found online at http://dx.doi.org/10.7717/peerj.10868#supplemental-information.

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
