# Peer review of "Genome size versus geographic range size in birds"

_PeerJ, doi:10.7717/peerj.10868_

## Round 0.1 · original submission · Major Revisions

I received detailed comments about your manuscript from three reviewers. Your paper should become acceptable for publication pending a major revision in light of the reviewers' comments. Please, follow all suggestions and send me your revised manuscript as soon as you can.

·

Basic reporting

1. As more avian genomes become available, new opportunities arise for analyzing global patterns of genetic factors underlying the geographical variation in diversity of life on Earth. This paper examines how genome size (as available on a genome size database) correlates with how widespread species are, and also examines the relationship with number of chromosomes, body size and brain-size (as proxy for behavioural flexibility). This is a novel and interesting topic, but in view of the many historical and ecological factors that affect range-size, the interpretation of possible correlations may be challenging. The paper is clear and well formulated, in professional English.

Experimental design

2. The introduction provides a clear logical framework, and the structure of the MS, and logics of the analysis, is fine, over all. I assume that brain-size (in various places in the text) refers to residual brain-size. The figures are relevant and clear. I could not see to what extent the raw data is made available

Validity of the findings

3. While I see no problems with the data analysis that was done, the authors are missing some important macroecological facts and have overlooked some confounding factors. These need to be taken into account when interpreting the data.
One premise of the study is that birds can fly, and therefore should be little affected by physical barriers (lines 65-66). This is true for high-latitude environments, but note that only few avian lineages are widely distributed (thus, only 5 of 145 families of passerine birds are represented on all continents). Most lineages have diversified only within a rather restricted part of the land surface, and many tropical species are (albeit able to fly) highly reluctant to cross unfamiliar habitat. The dispersal of birds are highly dependent on their life stragegies of the particular lineages! With respect to the ability to adapt to a broad range of climates, birds are often assumed to be constrained by niche conservatism. But it is important to note that niche conservatism appears to be assymetrical: ancient tropical groups cannot easily adapt to, and expand into, cold/harsh climates, but groups that evolved at high latitudes and are able to tolerate cold are actually thermally flexible and easily adapt to new climates, and are often founders of breeding populations (and new proliferations of species) within the tropics (e.g., Smith et al. Ecol. Letters 15(2012): 1218-25; Khaliq et al. J. Biogeogr. 42(2015): 2187-96; and see also Winger et al. in Biol. Reviews 94(2018): 737-753). For these reasons, the relationships that this study sets out to explore will probably differ greatly between northern and tropical (+austral) biota. The result may therefore depend much on the geographical sampling, and in the interpretation of the results, it is essential that latitudinal differences are taken into account and also that attention is given to how well the sample of species (with genome data) represents the global avian fauna.
The authors refer much to the studies of range-size variation by Kevin Gaston, some 20 years ago. Gaston’s papers used range-size data of exceedingly poor quality (very generalized range-maps from a fieldguide), which means that these studies greatly underestimated how many species that have really tiny distributions (and even the data from BirdLife provide rather crude range-size estimates). One of the great challenges in recent studies of macroecological patterns has been how to explain the highly aggregated geographical distribution of species with tiny geographical ranges in certain tropical areas (oroclines or locations near the margins of continental plates, and in archipelagos (see, for instance Rahbek et al. in Proc. Royal Soc. B 274(2007): 165-174 and Science 365(2019): 1108-1113, and for a review of diversification in archipelagos see Jønsson et al. in Ann. Rev. Evol. Evol. Syst. 48(2017): 231-253). The 50% of avian species with the smallest geographical ranges are found almost entirely at <30 degrees latitude (resulting in very high residual values in comparison with ecosystem productivity in certain areas). Unfortunately, the 50% of avian species with the smallest geographical ranges are very poorly represented in the genome data (thus, only one of the 65 species in Fig. 1B! [Serinus canaria]). In other words: this study is highly biased towards species with large to moderately small geographical distributions, and seriously neglects the large number of tropical species with really tiny distributions (this bias probably explains the negative association between range size and body size). These are of two kinds: some old relict species, and paraspecies within some lineages that underwent rapid recent radiation within certain oroclines and archipelagos.

Additional comments

4. As long as small-ranged species is so strongly underrepresented in the genome dataset, there is not much to do about this bias (other than waiting for more genomes to be available, especially from the b10k project), but it must at least me mentioned as a shortcoming of the study, and it would probably help the interpretation if tropical species (centroid of range at <30 degrees lat.) are conspicuously marked with colour symbol in Fig. 4. Another bias (or complicating factor) may be that high flying powers (aerial foraging) has already been found to correlate with small genome size (Zhang et al. 2014), but flying powers also affects range-size.

·

Basic reporting

In my view, the authors have to work better the argument about the bigger the genome, the better adapted an organism is to deal with multiple and more diverse environmental pressures. Put it another way, is there previous evidence that at least some sort of selection could lead to bigger genome sizes? They do not mention any empirical evidence in that direction.

Also, If they are stating the hypothesis of a positive correlation between genome size and geographic range for the first time, then they have to make it explicit. If not, then cite other works which have explicitly discussed that genome size co-varies with range size...if they are making this claim for the first time, then it is important to think about at least one alternative hypothesis / outcome, such as genome regulation being more important to adaptation that genome size per se. I am sure the literature is full of discussions like that, but the authors did not mention then here.

Experimental design

It is worth mentioning that the topology of the tree upon which their best explanatory model was based (Fig. 1B), is very different from the "best" trees achieved with more powerful genomic datasets published recently (see Jarvis et al. 2014: 10.1126/science.1253451)....so, we know the topology of Fig. 1B is likely inaccurate, which raises the issue of whether the correlations found were spurious, particularly considering the very low explanatory power of the models selected, and the slight strength of the correlation observed.

I did not check how the topology of Fig. 1A, compares with that of Jarvis et al. 2014, but Jetz et al. phylogeny is indeed fraught with problems and cannot be compared with estimates based on more powerful genomic datasets.

Validity of the findings

The models with better explanatory power still accounted to a maximum of 17% of the total variation, which is still very little, particularly considering that this was achieved with the smallest dataset, which was the least representative for the entire Aves class. Again, this raises the issue of whether the correlations found were spurious, particularly considering the very low explanatory power of the models selected, and the slight strength of the correlation observed between genome size and geographic range.

Additional comments

I have made several detailed comments directly onto the text which I hope will be useful to the authors.

·

Basic reporting

The manuscript “Genome size and geographic range size in birds” by Grzywacz and Skórka addresses an interesting and potentially important question that links the genome to biogeography. I would like to start by disclosing that I would not have undertaken this study because I would have predicted that it would almost certainly yield a null result. Since unexpected results are intrinsically exciting, I think this paper could have a high impact on the field and prove to be influential. However, this excitement regarding the results is contingent on the results being correct; I am not (yet) convinced that this is the case.

I would like to break my review of the manuscript into two major sections:

1. The nature of the question (and the authors framing of the question).

I have two reasons why I would have expected a null outcome from this study: 1) there is simply not that much variation in avian genome sizes; and 2) there is actually a lot of variation in avian karyotypes. In other words, when considered relative to most tetrapod groups almost all birds have the same amount of DNA per haploid genome but its organization into chromosomes varies quite a bit (in large part reflecting the large number of microchromosomes in birds). I expect the impact of genome size on whole organism phenotypes to be subtle (which, to be fair, is precisely what the authors find) so I would have expected the minimal variation in genome size to be swamped out by other factors. On the other hand, I would have expected to high degree of variation in chromosome number to simply lead to noise. This is actually why I find the study interesting!

With that said, the authors really do not acknowledge the limited variation in avian. genome sizes, something readily apparent from, for example, the Andrews et al. (2009) paper they cite (cf. Fig. 2 of Andrews et al., note that the scale of that figure is from 1.15 pg of DNA per haploid genome to 1.45 pg – i.e., not much variation). In the second paragraph, the authors cite a paper discussing genome size-phenotype linkages that highlights dinoflagellates (Hou and Lin 2009), a group that is known for wide variation in genome size and unusual chromosomal organization (relative to other eukaryotes). I am not opposed to citing Hou and Lin. Indeed, “unusual” taxa sometimes emphasize general patterns that may be much more subtle in less unusual taxa. But the authors should acknowledge the nature of the variation in genome size.

In practical terms, the authors should provide a figure similar to Fig. 2 of Andrews et al. (2009), updated with the new data they have incorporated. They should also include the chromosome number information they used and, ideally, this would be plotted on a large-scale phylogeny. An excellent example of such a figure can be found in Fig. 6 of Kretschmer et al. (2018). Although I don’t want to be too specific, I would not recommend a tree with all tips; I think it would be a more readable figure would collapse groups down to major lineages (e.g., to orders) and put a bar charts with error bars for the standard deviation (or interquartile range) or just numbers adjacent to the tips. This would give readers an idea of the scale of the variation in genome size within birds.

With all of that said, I’d like to applaud the authors for trying this. If true, I’m very interested in the effect they see. Indeed, the authors make the point that “…the explained variation is low, thus statistical significance does not necessary[sic] mean biological meaning” (their lines 198-199). Given the limited variation in genome size I would say that that finding any relationship at all – even a weak one – has the potential to be interesting and meaningful.


2. Inadequate estimates of phylogeny used for comparative methods

Adequate phylogenetic controls are critical for this project, but the tree they use is poorly documented and very old. Specifically, they state that they use the Jetz et al. (2012) tree, but the Jetz tree is actually two different sets of trees, one based on constraints from Ericson et al. (2006) and one based on constraints from Hackett et al. (2008). It appears to me (based on the authors Fig. 1) that the authors used the Ericson constraint set. I strongly recommend that the authors use the Hackett tree set; this is what most users of the Jetz tree use. Moreover, the Hackett et al. (2008). tree was based on an analysis of 19 loci whereas the Ericson et al. (2006) tree was based on five loci (and the constraints Jetz used were actually based on one of the supporting figures in Ericson et al. that included a mere four loci). Regardless, it is critical that the authors are clear regarding the trees they used.

With all of that said, the Jetz tree is now showing its age. The analysis Jetz and colleagues used was constrained by the Ericson and Hackett “backbone” trees and by taxonomy. The use of taxonomy allowed Jetz et al. to include taxa for which there was no actual data, but that practice led to some very problematic results (e.g., Wang et al 2017). Moreover, there are parts of the Jetz topology for taxa with no data that have 100% support (Hosner et al. 2015; this probably reflects the difficulty of running the MCMC chain long enough to adequately sample the posterior distribution when so many taxa are included). I am not opposed to including analyses using the Jetz tree because it is still the only really easy to use avian megaphylogeny, though it should be used with caution.

I stated that the Jetz tree is the only really easy to use avian megaphylogeny, but there is now a “replacement” that isn’t that hard to use: the tree used by Ksepka et al. (2020). It is based on a reanalysis of the Burleigh et al. (2015) supermatrix with constraints from the Jarvis et al. (2014) tree, which reflects analyses of about 40 Mbp of aligned data and includes more than 10,000 loci. The tree is available as supporting data from the publication, although it may be a bit of a pain to use. Specifically, there is a dated version trimmed to a subset of taxa (supplementary file S5 of Ksepka et al.) and a version without rate smoothing that can be converted to an ultrametric form (supplementary file S4, which it has fossil calibrations readable by the program r8s [available from http://ceiba.biosci.arizona.edu/r8s/index.html] included in the file but the tree needs to be converted to an ultrametric format). I think that, at a minimum, the authors should see how many of their taxa overlap with the file S5 tree and conduct an analysis of their data using that tree (after trimming to their taxa). This is why I could see retaining analyses based on Jetz et al. – the authors could do an analysis using the Jetz tree and all taxa and then trim down to taxa that overlap with the Ksepka file S5 tree and do an analysis using fewer taxa and that tree (or course, this would be irrelevant if all of their taxa are present in the Ksepka file S5 tree).

I wish there was a truly up to date avian megaphylogeny with dates out there; such a tree does not exist at this point. Unfortunately, the Jetz tree is definitely showing its age at this point and the Ericson backbone (14 years old at this point) is definitely problematic.

A final comment about presentation of trees: the authors clearly use an older taxonomy in their figure 1. They should update their ordinal names based on a modern taxonomy (i.e., the IOC World bird list; Gill et al. 2020). The use of the much older circumscription of e.g., Falconiformes serves to obscure the tree. Even in the early large-scale bird trees of Ericson et al. and Hackett et al. it was clear that Accipitriformes and Falconiformes sensu stricto (i.e., limiting extant Falconiformes to Falconidae) were not a clade. This is only amplified by more modern trees like Jarvis et al. Using Falconiformes in the old-fashioned manner obscures rather than illuminates.

And now one final comment about the comparative methods: the authors state that “…non-linear methods do not include phylogenetic relations among species” (lines 202-203) of their manuscript. This is not strictly true, Quader et al. (2004) discuss the issue of non-linearity. This paragraph is an excellent one, but it should be tweaked to incorporate Quader et al. I think the bigger point that there is no easy way to incorporate non-linearity is true. On the other hand, I’m not convince that is the biggest limitation of available methods.

Some minor issues:

The authors generally use excellent English but there are a few glitches. For example, I used [sic] above and the authors should change “necessary” to “necessarily” in that case. I feel a little bad for even saying this because it is always hard to write in a language other than your native tongue and I would not be able to do anywhere as well as the authors if I were writing in a language other than English. However, the authors should be conscious regarding the issue of language and they might consider asking a native English-speaker colleague to help them with a read through. In that context, they begin some paragraphs with “Interestingly, …” I dislike this, since it does not highlight anything in particular. After all, everything in a paper should make the bar of being interesting enough to include! Unless a sentence begins either with a contrasting or with an amplifying word/phrase (e.g., “In contrast,” “In fact,” etc.) it is probably best to avoid constructions similar to “Interestingly, …”. This is even more true for the beginnings of paragraphs.

Finally, I’m not sure what to make of the authors’ finding that “…body mass in the multivariate models and path analyses was negatively correlated with the range size” (their lines 183-184). I like that they point this out. Perhaps a little expansion of the reasons why this is not expected are warranted.


Disclosures:

A general policy for PeerJ is to make sure authors are aware that they should not be pushed into adding citations for papers written by reviewers or editors. I think this is a good policy. Therefore, I would like to disclose that I am a co-author on several papers cited below (e.g., Hackett et al., Jarvis et al., Ksepka et al.). In some cases, I think citation of those works is critical; for example, I think using the Ksepka et al. tree is really important. In other cases, my citation within this review was merely to illustrate a point (e.g., Hosner et al., Wang et al. illustrate issues with Jetz et al.) and I certainly don’t want to push for citations of my work beyond that which is appropriate given standard scientific ethics. In that context, I do not expect everything I cited in this review to be cited by the authors (moreover, some of the papers I cite are already cited by the authors).

I also want to say that like to sign reviews, regardless of whether they are positive or negative. I feel signing reviews helps the reviewer stay focused on providing constructive feedback. My overall evaluation of this manuscript is positive – I will say that “I want to believe” that the authors have found the relationship they say they have found. Of course, I also want them to convince me (and other readers!) that their results are robust. They’ve already convinced me that their results are interesting.

Edward L. Braun
Professor of Biology
University of Florida
Gainesville, FL USA

References for this review:

Andrews CB, Mackenzie SA, Gregory TR. 2009. Genome size and wing parameters in passerine birds. Proceedings of the Royal Society B. 276: 55–61 DOI:10.1098/rspb.2008.1012.

Burleigh, J. G., Kimball, R. T., & Braun, E. L. (2015). Building the avian tree of life using a large-scale, sparse supermatrix. Molecular phylogenetics and evolution, 84, 53-63.

Ericson, P. G. P., Anderson, C. L., Britton, T., Elzanowski, A., Johansson, U. S., Källersjö, M., et al. (2006). Diversification of Neoaves: integration of molecular sequence data and fossils. Biology letters, 2(4), 543-547.

Gill F, D Donsker & P Rasmussen (Eds). 2020. IOC World Bird List (v10.2). doi: 10.14344/IOC.ML.10.2. https://www.worldbirdnames.org/

Hackett, S. J., Kimball, R. T., Reddy, S., Bowie, R. C., Braun, E. L., Braun, M. J., et al. (2008). A phylogenomic study of birds reveals their evolutionary history. Science, 320(5884), 1763-1768.

Hosner, P. A., Braun, E. L., & Kimball, R. T. (2015). Land connectivity changes and global cooling shaped the colonization history and diversification of New World quail (Aves: Galliformes: Odontophoridae). Journal of Biogeography, 42(10), 1883-1895.

Hou Y, Lin S. 2009. Distinct Gene Number-Genome Size Relationships for Eukaryotes and Non- Eukaryotes: Gene Content Estimation for Dinoflagellate Genomes. PLoS ONE 4(9): e697 DOI:10.1371/journal.pone.0006978.

Jarvis, E. D., Mirarab, S., Aberer, A. J., Li, B., Houde, P., Li, C., et al. (2014). Whole-genome analyses resolve early branches in the tree of life of modern birds. Science, 346(6215), 1320-1331.

Jetz W, Thomas GH, Joy JB, Hartmann K, Mooers AO. 2012. The global diversity of birds in space and time. Nature 491: 444–448 DOI:10.1038/nature11631.

Ksepka, D. T., Balanoff, A. M., Smith, N. A., Bever, G. S., Bhullar, B. A. S., Bourdon, E., et al. Tempo and Pattern of Avian Brain Size Evolution. Current Biology.
volume 30, issue 11: 8 June 2020, Pages 2026-2036.e3

Kretschmer, R.; Ferguson-Smith, M.A.; De Oliveira, E.H.C. Karyotype Evolution in Birds: From Conventional Staining to Chromosome Painting. Genes 2018, 9, 181.

Quader, S., Isvaran, K., Hale, R. E., Miner, B. G., & Seavy, N. E. (2004). Nonlinear relationships and phylogenetically independent contrasts. Journal of Evolutionary Biology, 17(3), 709-715.

Wang, N., Kimball, R. T., Braun, E. L., Liang, B., & Zhang, Z. (2017). Ancestral range reconstruction of Galliformes: the effects of topology and taxon sampling. Journal of Biogeography, 44(1), 122-135.

Experimental design

As detailed in my complete review (the "basic reporting"), my primary concern with the experimental design is the use of a older phylogeny for their comparative methods.

Validity of the findings

As detailed in my complete review (the "basic reporting"), I think the findings could be valid but I feel the authors like to do a bit more to present a convincing case

---

## Round 0.2 · Minor Revisions

I received detailed comments about your updated manuscript from three reviewers. All of them are happy with the changes that you made and that improved the manuscript significantly. However, there are still a few issues to be worked on. Your paper should become acceptable for publication pending a minor revision in light of the reviewers' comments. Please, follow all suggestions and send me your revised manuscript as soon as you can.

·

Basic reporting

The paper has been successfully revised following the first review. The work is well presented, in clear English, and has a clear focus and interesting results

Experimental design

By using two alternative sets of phylogenetic data and analysis a much more robust result has been obtained. The analyses are well done and I see no weak points.

Validity of the findings

The results, as presented in lines 214-237, are well supported and presented

Additional comments

The Authors have followed well the comments by reviewers, especially by taking into account other phylogenetic data (Ksepka et al.) and by pointing out various confounding factors and the undersampling of species with very localized distributions (lines 290-323). The MS is now much more robust and well formulated, and I have only a few minor extra points:
The aspect of how avian flight has affected genome size is mentioned on line 247, but it seems relevant also to mention in the Introduction (maybe after line 90), and in line 302) how avian flight has required massive changes (for light weight and increased energy efficiency) of all aspects, including the size of the genome. I assume that Zhang et al. 2014 would be a useful reference here.
Further details: In line 40, I think it would be better to change ‘may’ to ‘could’ (twice).
Lines 275-277: Gaston used data from a field guide from his first studies of range-size variation, but actually the data source was different in the study of waterfowl by Gaston & Blackburn. But even here, the range size data are extremely crude as they used presence-absence data in a grid of 10x10 geographical degrees.

·

Basic reporting

All standards met.

Experimental design

All standards met.

Validity of the findings

All standards met.

Additional comments

The authors have made tremendous progress with this newer version of the manuscript and I am happy with what they have accomplished at this point: a somewhat daring, yet, sober (i.e., acknowledging all or most of the study´s limitations) analysis of the correlation between genome and ranges sizes, based on the best available evidence published so far. With that said, it is quite fortunate (or not!) that a recently published study based on 363 whole genomes covering from 92.4% of bird families (https://doi.org/10.1038/s41586-020-2873-9) offers quite a window of opportunity to be used in this paper as well. Interestingly, many of the shortcomings carefully addressed by the authors in this second version are due to problems that could be circumvented if this newer phylogeny had been used. For instance overall taxonomic coverage, as well as that of tropical taxa, is much better now at the genomic level. So,while I have no doubt that even a separate more "controlled" analysis focused just on this new tree would be a tremendous addition to this paper (while keeping the current presented results), making it probably even more cited, I realized that these analyses could potentially never end, given the wealth of data that is becoming available at a faster pace now (see also, this mind-boggling phylogenomic study focused at birds as well: https://academic.oup.com/mbe/advance-article/doi/10.1093/molbev/msaa191/5891114). So, while I am recommending acceptance as is, I would strongly recommend them to make an effort to include at least the whole genome more complete tree that just came out. I realize it is a lot of additional work, but in my view it would just add some more value to their contribution.

·

Basic reporting

I feel this version of the manuscript is improved and is now more than adequate.

Experimental design

I feel the analysis is rigorous. There are intrinsic limitations to the data the authors analyze, which they describe in a section called "Study limitations". I like this quite a bit. Overcoming these limitations (e.g., geographic biases; the problems remaining in avian phylogenies) would require another decade of data accumulation (e.g., more information from tropical species, better megaphylogenies). This is an interesting story and will push the field forward; the authors should be able to publish this now and not wait for "perfect data".

Validity of the findings

I believe the findings are valid. I provide additional details above in section 2 "Experimental design"

Additional comments

I really liked this revision. I feel that you carefully considered and addressed the my comments and the comments made by by the other reviewers.

I debated an accept vs minor revisions recommendation. I have one minor recommendation related to Fig. 1. I really love the new Fig. 1 -- I believe that it conveys very important information to readers. However, I do have one minor criticism. The authors state "It is very difficult to locate these orders in the phylogenetic tree by Jarvis et al. (2014) or Ksepka et al. (2020). Therefore, we trimmed Jetz et al. (2012, 2014) tree and plotted summary statistics for bird orders."

I understand this and it is reasonable response. This figure is largely to orient readers rather than an effort to provide phylogenetic control. However, I think the authors should (minimally) change the Fig. 1 legend or (ideally, in my opinion) revise the tree. Specifically, the legend is:

"Figure 1
A consensus phylogenetic tree and studied variables
Bars are means with standard deviations. Sample size (number of species) is given next to each bar. NA indicates that data was not available"

At the absolute minimum it should acknowledge that this tree is the Jetz et al. tree.

However, I would like to point out that there is a modern tree that includes all avian orders and meets the criterion of being the result of analyses using phylogenomic datasets - it is Kimball et al. 2019. The Kimball et al. 2019 tree is a supertree - a meta-analysis of trees based on large-scale phylogenetic studies (most source trees were based on 1000's of loci). The Kimball et al. tree is (unsurprisingly) very similar to the Jarvis et al. 2014 tree where they overlap. It seems to me like the ideal solution for Fig. 1. I provide a full citation below. All orders that the authors show in their Fig. 1 are in the tree shown as Figure 3 of Kimball et al. 2019. In retrospect, I should have pointed the authors to this tree. Note also that Fig. 1 of their paper would be fine as a cladogram (i.e., a tree without meaningful branch lengths; in fact, I think a cladogram would be better than what they have).

Kimball, R.T.; Oliveros, C.H.; Wang, N.; White, N.D.; Barker, F.K.; Field, D.J.; Ksepka, D.T.; Chesser, R.T.; Moyle, R.G.; Braun, M.J.; Brumfield, R.T.; Faircloth, B.C.; Smith, B.T.; Braun, E.L. A Phylogenomic Supertree of Birds. Diversity 2019, 11, 109. https://doi.org/10.3390/d11070109

In the interest of full disclosure, I want to call the authors' (and editor's) attention to the fact that I am a co-author of Kimball et al. 2019. I. recognize that this is a touchy issue and I try to avoid recommending a citation to my own work unless I feel it is absolutely critical. That is why I have provided an alternative in this review. With that said, I hope that the authors will carefully consider using the Kimball et al. tree; it is a modern tree and I will stress that both I and one other review commented on the issues with the Jetz tree.

I really hope that this paper is ultimately accepted. In my opinion, it will be an important contribution to the literature.

Edward L. Braun
Professor of Biology, University of Florida

---

## Round 0.3 · accepted · Accept

Congratulations. I reviewed your responses to the reviewers, and I think your paper is ready to be published.